

# Integrating population genetic structure, microbiome, and pathogens presence data in *Dermacentor variabilis*

Paula Lado[1], Bo Luan[2], Michelle E.J. Allerdice[3], Christopher D. Paddock[3], Sandor E. Karpathy[3] and Hans Klompen[1]

[1] Evolution, Ecology, and Organismal Biology, The Ohio State University, Columbus, OH, United States of America
[2] Statistics, The Ohio State University, Columbus, OH, United States of America
[3] Rickettsial Zoonoses Branch, Centers for Disease Control and Prevention, Atlanta, GA, United States of America

## ABSTRACT

Tick-borne diseases (TBDs) continue to emerge and re-emerge in several regions of the world, highlighting the need for novel and effective control strategies. The development of effective strategies requires a better understanding of TBDs ecology, and given the complexity of these systems, interdisciplinary approaches are required. In recent years, the microbiome of vectors has received much attention, mainly because associations between native microbes and pathogens may provide a new promising path towards the disruption of pathogen transmission. However, we still do not fully understand how host genetics and environmental factors interact to shape the microbiome of organisms, or how pathogenic microorganisms affect the microbiome and vice versa. The integration of different lines of evidence may be the key to improve our understanding of TBDs ecology. In that context, we generated microbiome and pathogen presence data for *Dermacentor variabilis*, and integrated those data sets with population genetic data, and metadata for the same individual tick specimens. Clustering and multivariate statistical methods were used to combine, analyze, and visualize data sets. Interpretation of the results is challenging, likely due to the low levels of genetic diversity and the high abundance of a few taxa in the microbiome. *Francisella* was dominant in almost all ticks, regardless of geography or sex. Nevertheless, our results showed that, overall, ticks from different geographic regions differ in their microbiome composition. Additionally, DNA of *Rickettsia rhipicephali, R. montanensis, R. bellii,* and *Anaplasma* spp., was detected in *D. variabilis* specimens. This is the first study that successfully generated microbiome, population genetics, and pathogen presence data from the same individual ticks, and that attempted to combine the different lines of evidence. The approaches and pre-processing steps used can be applied to a variety of taxa, and help better understand ecological processes in biological systems.

Corresponding author
Paula Lado, ladohenaise.1@osu.edu, pau.parasito@gmail.com

## INTRODUCTION

As vector-borne diseases continue to emerge and re-emerge in several regions of the world, there is an urgent need for a better understanding of their ecology, including the tripartite pathogen-vector-host relationship, to ultimately develop effective control strategies. Ticks represent an increasing threat to One Health given the range expansion of some species, and the frequent identification of new tick-borne pathogens (*Paddock et al., 2016*; *Eisen et al., 2017*). Due to the complex nature of diseases transmitted by ticks, their control requires interdisciplinary studies and collaboration (*Murgia et al., 2019*).

In recent years, the microbiome (referring to bacteria only throughout the manuscript) of vectors (mostly mosquitoes and ticks) has received much attention. The main reason is that associations between native microbes and pathogens may provide a new promising path towards the disruption of pathogen transmission (*Narasimhan & Fikrig, 2015*; *Bonnet et al., 2017*). In the case of ticks, microbes can be acquired from the environment (e.g., through the spiracles), horizontally (from the host, during blood feeding), or vertically (maternally inherited). Relationships between the tick and a particular microbe or set of microbes can range from mutualistic to parasitic (*Casadevall, Fang & Pirofski, 2011*; *Bonnet et al., 2017*). Microbes can affect tick fitness, vector competence, and pathogen transmission (*Bonnet et al., 2017*; *Budachetri et al., 2018*). For example, a few maternally inherited microbes have been proposed as primary symbionts necessary for tick success by providing vitamins that are lacking in blood (*Smith et al., 2015b*; *Duron et al., 2017*; *Guizzo et al., 2017*). Without these microbes, the ticks' fitness is negatively impacted (*Zhong, Jasinskas & Barbour, 2007*). Other symbionts interact with pathogenic microorganisms, either positively or negatively. High proportions of the endosymbiotic *Rickettsia bellii* inhibit the transmission of the pathogenic *Anaplasma marginale* (*Gall et al., 2016*), and *R. parkeri* is likely excluded from *A. maculatum* by Candidatus *Rickettsia andeanae* (*Paddock et al., 2015*), presumably through a phenomenon known as interference. On the other hand, cases of facilitation have also been reported; such as a positive relationship between the proportion of the microbiome occupied by *Francisella* endosymbiont (FLE) and the infection level of *F. novicida* (*Gall et al., 2016*).

Even though a considerable amount of information and knowledge has been generated during the last decade or so, microbiome research on ticks is still in its infancy. For example, we still do not fully understand how host genetics and environmental factors interact to shape the microbiome of organisms (*Spor, Koren & Ley, 2011*; *Goodrich et al., 2014*; *Steury et al., 2019*), or how pathogenic microorganisms affect the microbiome and vice versa. Integration of different lines of evidence may be the key to improve our understanding of TBDs ecology. As highlighted by *Griffiths et al. (2018)*, research exploring the links between, for example, vector genetics, microbiome composition and structure, and pathogen susceptibility may enable a better understanding of the factors governing disease in vulnerable populations. Unfortunately, to the best of our knowledge, there are no such studies on ticks. There are a few studies based on vertebrate taxa that have attempted to incorporate genomic and microbial data with data on environmental variation (*Fietz et al., 2018*; *Griffiths et al., 2018*; *Steury et al., 2019*). Overall, these studies have shown that

genetically divergent host populations, exhibited more divergent microbiomes (*Smith et al., 2015a*; *Griffiths et al., 2018*; *Steury et al., 2019*). In amphibians, *Griffiths et al. (2018)* found that the genetic distance among hosts was correlated with microbial community dissimilarity when controlling for geographic distance. Nevertheless, the same study showed that at the site-level the microbiome did not mirror the host population genetic structure. *Steury et al. (2019)* investigated populations of Threespine Stickleback and concluded that the microbiome composition was better predicted by fish population genetic divergence than by geographic distance and environment. It is worth noting that the global trends appear to be driven by a subset of the microbiome. In other words, the influence of host genetic factors on the microbiome composition depends on the bacterial taxa in question; host genetics may affect the presence of some microbes, but presence of others may be better explained by, for example, the environment (*Fietz et al., 2018*). In the same way, differences between populations could be the result of differences in the relative abundance of a small subset of microorganisms (*Steury et al., 2019*).

The vertebrate results suggest that integrating microbiome and population genetic data may lead to better understanding of the ecology of tick-borne diseases. Do ticks from different genetic clusters harbor a different microbiome? Do infected ticks belong to a specific genetic cluster? Is the microbiome of infected ticks different from that of non-infected ticks? To start answering these questions, it is paramount to first generate high quality data, and then to develop an appropriated framework to integrate the different lines of evidence. This exploratory study focuses on *Dermacentor variabilis* (Say), a North American tick species that commonly bites humans, and that historically has been implicated in the transmission of several pathogens (*Hecht et al., 2019*). *Dermacentor variabilis* is also one of the most widely distributed ticks in the United States (*Eisen et al., 2017*), one for which population genetic data was already available. During this investigation, we first generated high-quality microbiome data and determined the presence/absence of several pathogens. Next, we combined those data sets with an existing population genetics data set for *D. variabilis* (see *Lado et al., 2019*). All three pieces of data (microbiome, pathogens, and population genetics) were derived from each of the individual ticks included in this study. The goal of this preliminary study was to integrate the different lines of evidence for *D. variabilis* ticks using clustering methods and multivariate statistics to identify trends and patterns of variation. Our expectation was that *D. variabilis* ticks that are genetically more alike (i.e., that belong to the same genetic cluster), will also be more alike at the microbiome level. Additionally, we hypothesized that infected and non-infected ticks would have a subset of microbes in their microbiome that are characteristic for each category.

## METHODS

### Tick samples

The tick specimens employed in this study are the same individual ticks from *Lado et al. (2019)*, with the exception of two specimens for which we did not have enough genetic material. The sample includes 64 adult *D. variabilis* collected from California ($n = 3$),

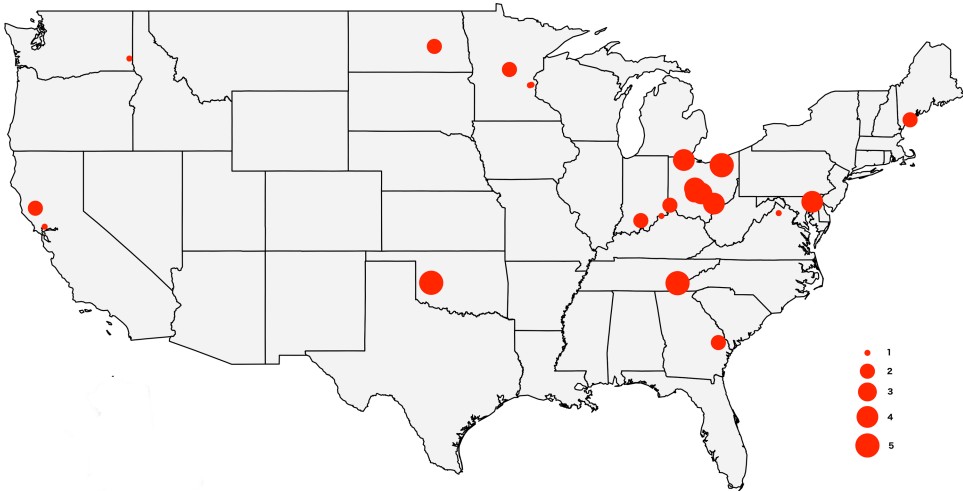

**Figure 1** **Map of the USA showing the sampling locations.** The size of the circles reflects the number of specimens analyzed from each locality, from 1 to 5.

Georgia ($n = 2$), Indiana ($n = 3$), Maryland ($n = 4$), Maine ($n = 3$), Minnesota ($n = 5$), North Dakota ($n = 1$), Ohio ($n = 31$), Oklahoma ($n = 5$), Tennessee ($n = 5$), Virginia ($n = 1$), and Washington ($n = 1$) (Fig. 1). All specimens were wild caught by dragging a 1m ×0.8m cloth thorugh the vegetation, and preserved in 95% ethanol. We collected ticks from vegetation (unfed) to minimize the amount of genetic material from the host. Collection data for the specimens used in this study can be accessed through the Ohio State Acarology Collection (OSAL) online database (https://acarology.osu.edu). Metro Parks permitted collection of tick species throughout the Metro Parks. The Division of State Parks (DNR), Indiana Department of Natural Resources authorized tick collection at Brown County State Park, Clifty Falls State Park, O'Bannon Woods State Park, Monroe Lake, Splinter Ridge Fish and Wildlife Area, Clark State Forest, and Ferdinand State Forest.

## DNA extraction
Before DNA extraction the ticks were surface sterilized following *Lado et al. (2018)*; specifically, the washes consist on following this procedure twice: commercial bleach 3% for one minute, washed in distilled water, and 95% ethanol for another minute. DNA extractions of individual tick specimens were performed using the QIAgen Blood and Tissue kit following the manufacturers' instructions, with one exception as in *Lado et al. (2018)*: during the incubation in ATL buffer the posterior-lateral part of the tick idiosoma was cut with a scalpel to allow a better penetration to the buffer to the tick's tissues. The cuticle of all ticks was recovered and kept as voucher. OSAL accession numbers are listed in Table 1. Genomic DNA was quantified using a Qubit 3.0 fluorometer; and then aliquoted and kept in the freezer until used.

## Population genetic data
For the integrative portion of the analysis, we employed all genetic data generated previously, and followed the genetic clusters nomenclature from *Lado et al. (2019)*. Details

**Table 1 General information of *Dermacentor variabilis* samples used in this study.** All individuals are from the USA. Each row corresponds to an individual tick specimen and the columns to the collection information.

| Raw data ID | Voucher ID | Collection event ID | Sex | US state | Locality | Coordinates |
|---|---|---|---|---|---|---|
| 119592-6 | OSAL119943 | OSAL119592 | Female | IN | Hoosier National Forest | 38.52, −86.44 |
| 119592-7 | OSAL119944 | OSAL119592 | Male | IN | Hoosier National Forest | 38.52, −86.44 |
| 119273-2 | OSAL119553 | OSAL119273 | Female | IN | Splinter Ridge Wildlife area | 38.75, −85.20 |
| 119241-2 | OSAL119260 | OSAL119241 | Female | OH | Battelle Darby Creek MP | 39.9, −83.21 |
| 119241-3 | OSAL119261 | OSAL119241 | Male | OH | Battelle Darby Creek MP | 39.9, −83.21 |
| 119241-4 | OSAL119262 | OSAL119241 | Male | OH | Battelle Darby Creek MP | 39.9, −83.21 |
| 119243-1 | OSAL119267 | OSAL119243 | Female | OH | Glacier Ridge MP | 40.13, −83.18 |
| 119243-2 | OSAL119268 | OSAL119243 | Female | OH | Glacier Ridge MP | 40.13, −83.18 |
| 119243-3 | OSAL119269 | OSAL119243 | Male | OH | Glacier Ridge MP | 40.13, −83.18 |
| 119243-4 | OSAL119270 | OSAL119243 | Male | OH | Glacier Ridge MP | 40.13, −83.18 |
| 119244-1 | OSAL119400 | OSAL119244 | Female | OH | High banks MP | 40.15, −83.03 |
| 119244-4 | OSAL119403 | OSAL119244 | Male | OH | High banks MP | 40.15, −83.03 |
| 119247-1 | OSAL119404 | OSAL119247 | Female | OH | Pickerington Ponds MP | 39.88, −82.79 |
| 119247-4 | OSAL119407 | OSAL119247 | Male | OH | Pickerington Ponds MP | 39.88, −82.79 |
| 119248-1 | OSAL119392 | OSAL119248 | Female | OH | Pickerington Ponds MP | 39.88, −82.80 |
| 119248-4 | OSAL119395 | OSAL119248 | Male | OH | Pickerington Ponds MP | 39.88, −82.80 |
| 119250-2 | OSAL119397 | OSAL119250 | Female | OH | Sharon Woods MP | 40.11, −82.95 |
| 119250-4 | OSAL119399 | OSAL119250 | Male | OH | Sharon Woods MP | 40.11, −82.95 |
| 110559-1 | OSAL110503 | OSAL110559 | Female | OH | Fernald Preserve | 39.29, −84.69 |
| 110559-2 | OSAL110504 | OSAL110559 | Female | OH | Fernald Preserve | 39.29, −84.69 |
| 119600B | OSAL129714 | OSAL119600 | Male | OH | Roads intersection | 39.13, −84.79 |
| 115093A | OSAL129721 | OSAL115093 | Female | OH | Cuyahoga Valley | 41.289, −81.573 |
| 115093B | OSAL129722 | OSAL115093 | Female | OH | Cuyahoga Valley | 41.289, −81.573 |
| 115093C | OSAL129723 | OSAL115093 | Female | OH | Cuyahoga Valley | 41.289, −81.573 |
| 115093D | OSAL129724 | OSAL115093 | Male | OH | Cuyahoga Valley | 41.289, −81.573 |
| 115093E | OSAL129725 | OSAL115093 | Male | OH | Cuyahoga Valley | 41.289, −81.573 |
| 119928A | OSAL129691 | OSAL119928 | Female | OH | Strouds Run SP | 39.369, −82.042 |
| 119928B | OSAL129692 | OSAL119928 | Female | OH | Strouds Run SP | 39.369, −82.042 |
| 119928C | OSAL129693 | OSAL119928 | Male | OH | Strouds Run SP | 39.369, −82.042 |
| 119928D | OSAL129694 | OSAL119928 | Male | OH | Strouds Run SP | 39.369, −82.042 |
| 119572A | OSAL129695 | OSAL119572 | Female | OH | Oak Openings MP | 41.549, −83.854 |
| 119572B | OSAL129696 | OSAL119572 | Female | OH | Oak Openings MP | 41.549, −83.854 |
| 119572C | OSAL129697 | OSAL119572 | Male | OH | Oak Openings MP | 41.549, −83.854 |
| 119572D | OSAL129698 | OSAL119572 | Male | OH | Oak Openings MP | 41.549, −83.854 |
| 119567A | OSAL129707 | OSAL119567 | Female | TN | Knoxville | 35.390, −84.226 |
| 119567B | OSAL129708 | OSAL119567 | Female | TN | Knoxville | 35.390, −84.226 |
| 119567C | OSAL129709 | OSAL119567 | Female | TN | Knoxville | 35.390, −84.226 |
| 119567D | OSAL129710 | OSAL119567 | Male | TN | Knoxville | 35.390, −84.226 |
| 119567E | OSAL129712 | OSAL119567 | Male | TN | Knoxville | 35.390, −84.226 |

**Table 1** (*continued*)

| Raw data ID | Voucher ID | Collection event ID | Sex | US state | Locality | Coordinates |
|---|---|---|---|---|---|---|
| N8805A | OSAL129702 | USNMENT01358805 | Male | OK | Washita Co. | 35.411, −99.059 |
| N8805B | OSAL129703 | USNMENT01358805 | Male | OK | Washita Co. | 35.411, −99.059 |
| N8805C | OSAL129704 | USNMENT01358805 | Male | OK | Washita Co. | 35.411, −99.059 |
| N8806A | OSAL129705 | USNMENT01358806 | Female | OK | Washita Co. | 35.411, −99.059 |
| N8806B | OSAL129706 | USNMENT01358806 | Female | OK | Washita Co. | 35.411, −99.059 |
| N8464B | OSAL129701 | USNMENT01358464 | Male | VA | Warren Co. | 38.893, −78.14 |
| 119951 | OSAL129711 | OSAL119951 | Female | GA | Statesboro | 32.42, −81.77 |
| 119952A | OSAL129715 | OSAL119952 | Female | GA | Statesboro | 32.42, −81.77 |
| 115086 | OSAL129836 | OSAL115086 | Male | MN | Carlos Avery | 45.287, −93.122 |
| 115087A | OSAL129717 | OSAL115087 | Female | MN | Camp Ripley | 46.076, −94.349 |
| 115087B | OSAL129718 | OSAL115087 | Male | MN | Camp Ripley | 46.076, −94.349 |
| 119918A | OSAL129829 | OSAL119918 | Female | MN | Columbus | 45.31, −93.02 |
| 119918B | OSAL129830 | OSAL119918 | Female | MN | Stutsman Co. | 47.23, −98.87 |
| N128168B | OSAL129979 | USNMENT01358520 | Male | ND | Stutsman Co. | 47.23, −98.87 |
| 115139A | OSAL129834 | OSAL115139 | Female | ME | Crescent Beach | 43.56, −70.23 |
| 115139B | OSAL129835 | OSAL115139 | Male | ME | Crescent Beach | 43.56, −70.23 |
| 115140 | OSAL129833 | OSAL115140 | Female | ME | Unknown | Unknown |
| 119276-2 | OSAL129559 | OSAL119276 | Female | MD | Aberdeen Providing Ground | 39.46, −76.12 |
| 119276-3 | OSAL129560 | OSAL119276 | Female | MD | Aberdeen Providing Ground | 39.46, −76.12 |
| 119276-5 | OSAL129562 | OSAL119276 | Female | MD | Aberdeen Providing Ground | 39.46, −76.12 |
| 119276-6 | OSAL129563 | OSAL119276 | Female | MD | Aberdeen Providing Ground | 39.46, −76.12 |
| 115101 | OSAL129845 | OSAL115101 | Male | CA | Napa Co | 38.215, −122.33 |
| 115102A | OSAL129846 | OSAL115102 | Male | CA | Lake Co. | 39.139, −122.886 |
| 115102C | OSAL129848 | OSAL115102 | Male | CA | Lake Co. | 39.139, −122.886 |
| 115105B | OSAL129852 | OSAL115105 | Female | WA | Whitman Co. | 46.623, −117.228 |

on how the population genetics data set was generated and analyzed can be found elsewhere (*Lado et al., 2019*). In brief, ticks were assigned to three different genetic clusters according to their population genetic structure: a generally "western cluster" ($n = 4$), an "eastern cluster" ($n = 51$), and a "northern cluster" ($n = 9$). This clusters usually correspond to the locations where ticks were collected.

## Detection of Rickettsiales through PCR
### PCR screening

Tick DNAs were tested by real-time PCR to detect three genera that include pathogenic microorganisms: *Rickettsia*, *Anaplasma*, and *Ehrlichia*. All ticks were initially screened using two real-time assays: (1) a TaqMan Pan-*Rickettsia* assay, which amplifies a portion of the 23S gene for all *Rickettsia* species using primers PanR8-F and PanR8-R (*Kato et al., 2013*), and (2) an EvaGreen Anaplasmataceae assay that targets a portion of the 16S gene using primers ECHSYBR-F and ECHSYBR-R, amplifying all *Anaplasma* and *Ehrlichia* species (*Li et al., 2002*).

Positive tick samples for Pan-*Rickettsia* (23S real-time assay) where further screened to identify the *Rickettsia* species present. Samples were subjected to conventional semi-nested PCR targeting the *omp* A gene of all spotted fever group *Rickettsia* (*Regnery, Spruill &*

*Plikaytis, 1991*; *Eremeeva et al., 2006*). PCRs were performed using 1 µM of each primer (Rr190.70, Rr190.602, Rr190.701), 10 µL of Taq PCR Master Mix (QIAGEN), 2 µL of sample DNA in the primary reaction or 2 µl of the primary reaction product in the secondary reaction, and water to bring the final reaction volume to 20 µl. Positive samples were processed as described below to identify species present. We also performed a *R. bellii* specific TaqMan assay targeting *glt* A, the citrate synthase gene (*Hecht et al., 2016*) for all positive samples for Pan-*Rickettsia*.

All real-time PCRs were performed in duplicate on a BioRad CFX 96 thermal cycler using 4 µL of template DNA in a final reaction volume of 25 µL for the Pan-*Rickettsia* and 20 µL for both the Anaplasmataceae and *R. bellii*-specific assays. We considered samples positive if one of the duplicates had a cycle threshold (Ct) <40. Two sets of negative controls and one set of positive controls were included on each plate, where water was used as the negative non-template control and DNA from cultured *R. rickettsii*, *E. canis*, or an *R. bellii* plasmid were used as positive controls, depending on the assay (*Hecht et al., 2019*).

### Amplicon purification and sequencing
Amplicons from *omp* A semi-nested PCR were visualized on 1.5% agarose gels using ethidium bromide. Amplicons were extracted and purified using the Promega Wizard SV Gel and PCR Clean-up System (Promega, Madison, WI). Products were bidirectionally sequenced using a BigDye Terminator v3.1 kit on an ABI 3500 genetic analyzer (Applied BioSystems, Carlsbad, CA) and assembled using Geneious version 7.0.4. (http:// geneious.com, *Kearse et al., 2012*). The nucleotide BLAST tool of the NCBI GenBank database was employed to compare the amplicon sequences to those sequences available in the database. Positive amplicons from the Anaplasmataceae assay were also sequenced following the above-mentioned procedure.

## Microbiome
### 16SrDNA library preparation and amplicon sequencing.
Genomic DNA samples, along with two negative controls, were taken to a final concentration of 10 ng/µl, and shipped to Argonne National Laboratory for library preparation and sequencing following standard procedures. The two negative controls correspond to a extraction blank and library blank *sensu* (*Kim et al., 2017*). The primer pair 515F/926R (*Walters et al., 2016*) was employed to amplify the V4–V5 variable regions of the 16S rDNA gene, and then the amplicons were sequenced on a MiSeq illumina platform, paired-end 251bp reads. The obtained reads were demultiplexed using MiSeq Reporter.

### Quality filtering, OTU picking, taxonomic assignments, and diversity calculations
Data were initially filtered as previously described in *Lado et al. (2018)*. Specifically, after demultiplexing, Cutadapt (*Martin, 2011*) was used to do an initial quality filter of the reads (threshold Q10), and to trim the adaptors if they were present in the filtered reads. Once the reads passed the initial filters, the QIIME 1.9.1 (*Caporaso et al., 2010*) pipeline was employed to: assemble the reads; cluster the reads (97% threshold); and to assign taxonomy. Open reference OTUs (operational taxonomic units) picking using uclust, and
taxonomic assignment using the Greengenes (*DeSantis et al., 2006*) and Silva132 (*Quast et al., 2013*) data bases. An alignment of representative sequences was used as input to generate a tree in FastTree (*Price, Dehal & Arkin, 2009*); as well as to construct OTU tables for each of the taxonomic level (i.e genus, family). OTUs abundant in the negative controls and suspected as contaminants were removed in R using the *decontam* package (*Davis et al., 2018*).

## Statistical analysis: integrating individual data sets

OTU tables were imported into R, and finer filtering was performed. First, we eliminated all OTUs that appeared in no more than two samples. Then, with the reduced data, we further eliminated OTUs with less than 0.5% relative abundance in all ticks. The dimensions of the data were therefore substantially reduced. Metadata was then added to the tables, including coordinates, sex, genetic cluster they belong to (following *Lado et al., 2019*), and the presence of: *Rickettsia*, *Anaplasma* or *Ehrlichia* species for each tick specimen. Additionally, ticks were assigned to three geographic regions following *Lado et al. (2019)*: eastern, western, and northern.

### Clustering approach

To determine how samples are related to one another when integrating all variables, a hierarchical clustering approach was taken. In this analysis the variables included were the filtered OTU table, tick ID, sex, genetic cluster, location (as coordinates), and presence of *Ricketts* ia or Anaplasmataceae agents. Hierarchical clustering was performed in R and the number of clusters was set to three ($k = 3$), which is the number of genetic clusters. We employed Gower's distance (*Gower, 1971*) as the measure of dissimilarity between samples because it is compatible with mixed data types (quantitative, nominal, and binary variables). We visualized the clustering results through the dendrogram generated by the hierarchy and demonstrated sample organization in space by using the first two principal components of all the filtered OTUs and location coordinates.

As a further exploration, we evaluated how well our clustering matched with the genetic clusters by the concordance index (C-index), which is an internal validation measure of goodness of matching. We also employed methods to determine the optimal number of clusters from the data. This allowed the comparison between the predicted number of clusters ($k = 3$), and the number of clusters determined inherently from the data set. For this purpose, two methods were used: the average silhouette and the elbow method. The former method computes the average silhouette of observations for different values of $k$. The optimal number of clusters is the one that maximizes the average silhouette over a range of possible values for k (*Kaufman & Rousseeuw, 1990*). On the other hand, the elbow method looks at the total within-cluster sum of squares (wss) as a function of the number of clusters. The optimal $k$ will be the smallest one such that adding another cluster doesn't improve much better the total wss.

### Ordination methods

Once the data was integrated, ordination methods were used to visualize the samples in space. We used both non-metric multidimensional scaling (NMDS) and Principal

coordinates analyses (PCoA) to determine if there was any patterning within the data. Both techniques reduce the high dimensional data into a two-dimensional representation. All analysis and graphs were done in R, employing the *phyloseq* package (*McMurdie & Holmes, 2013*).

### Test for significance for different variables

We used Welch's $t$-test to detect if the relative abundance of a specific species of microorganism differed across host sex or between genetic clusters. Such tests were shown to be flexible and robust even when samples sizes were unbalanced and group variances were unequal (*Delacre, Lakens & Leys, 2017*).

To compare microbial beta diversity across treatments (sex, genetic cluster, and region) we calculated the distance between microbial communities by using two metrics. The first one, Bray-Curtis distance, considers the relative abundance of bacteria, while the second, Jaccard's distance (or dissimilarity index), measures differences in presence/absence of Bacteria. All distances were calculated using the *vegan* package (*Oksanen et al., 2019*) in R. We then partitioned the matrices by treatment and performed permutational multivariate analysis of variance (PERMANOVA) (*Anderson, 2001*) to determine if beta diversity differed between treatments. PERMANOVA calculations were performed using the *adonis* function in the vegan package with 999 permutations. For those treatments that included more than two groups, a post hoc test was performed to identify the pairs of groups between which the bulk of differences occurred. For that purpose, we employed the *paiwise.adonis* function from the same package.

## RESULTS

### Detection of Rickettsiales through PCR

DNA belonging to the genus *Rickettsia* was detected in 15.6% (10/64) of the screened ticks including *R. montanensis* ($n = 6$, 9.3%); *R. rhipicephali* ($n = 2$, 3.1%); and *R. bellii* ($n = 2$, 3.1%). *Rickettsia bellii* was detected only in specimens collected from the western U.S.

*Anaplasma* DNA was found in a single tick (1.6%), collected in Oklahoma. The DNA sequence obtained was identified as "Uncultured *Anaplasma* sp. clone 15-3642 16S ribosomal RNA gene, GenBank number MG429812". The sequence generated in this study was 100% identical to MG429812, and the query coverage was also 100%, corresponding to bases 23 to 168.

None of the ticks were infected with *Ehrlichia* species. A summary of PCR results and metadata for each tick specimen is presented in Table 2.

### Microbiome

Microbiome data were successfully generated for all ticks. However, some issues were encountered when attempting to merge forward and reverse reads (only a minimal fraction of the reads merged correctly). Because of that and given the large amount of data obtained, only forward reads (R1) were employed for all downstream analysis. It is possible that the difficulties encountered when merging the reads is due to the primers pair selected, as other research groups have faced similar issues with these primers for different taxa (Argonne

**Table 2  Rickettsia spp. and Anaplasma spp. screening results divided by state.**

| State | Year of collection | Number of ticks positive for rickettsiales | Total number of ticks positive | | | |
|---|---|---|---|---|---|---|
| | | | *R. rhipicephali* | *R. montanensis* | *R. bellii* | *Anaplasma* spp. |
| California | 2017 | 2/3 (66.7%) | | | + (2) | |
| Georgia | 2013, 2016 | 1/2 (50%) | | + (1) | | |
| Indiana | 2017 | 0/3 | | | | |
| Maryland | 2017 | 0/4 | | | | |
| Maine | 2017 | 0/3 | | | | |
| Minnesota | 2017 | 0/5 | | | | |
| North Dakota | 2017 | 1/1 (100%) | | + (1) | | |
| Ohio | 2016, 2017 | 1/31 (3.2%) | | + (1) | | |
| Oklahoma | 2017 | 3/5 (60%) | + (2) | | | + (1) |
| Tennessee | 2017 | 3/5 (60%) | | + (3) | | |
| Virginia | 2017 | 0/1 | | | | |
| Washington | 2017 | 0/1 | | | | |
| Total | | 11/64 (17.2%) | 2 | 6 | 2 | 1 |

National Lab personal communication). The number of Illumina forward (R1) reads obtained was 2,131,680. After quality filtering the number of reads retained was 2,129,331 with an average length of 251 bp. The average number of sequences per sample was 32,795 (minimum = 9,331; maximum = 44,533). The number of reads for the extraction blank was 2349, considerably lower than all samples; and no reads were generated from the library blank.

A total of 41 orders of bacteria were identified (13 remaining after filtering), and three of them were dominant: Rickettsiales, Legionellales, and Enterobacteriales (Fig. 2A). At the genus level, 61 taxa were detected (File S1). Five taxa were removed since their relative abundance was greater in the negative control than in tick samples. Of the 56 genera remaining, 36 were eliminated during the filtering steps according to minimum relative abundance and the number of samples (see Methods section for details). Thus, 20 genera were kept for downstream analyses (File S1). Three taxa were highly abundant at the genus level: *Rickettsia* spp., *Francisella* spp., and Enterobacteriaceae "other". (Fig. 2B). *Francisella spp.* were present in all ticks, with a relative abundance ranging from 0.4 to 100% (mean = 81%, median = 98.5%). *Rickettsia* was detected in 12 (18.8%) of the ticks. In all cases where *Rickettsia* spp. had a relative abundance of at least 0.5% by microbiome analysis, tick specimens were also PCR positive, thus microbiome and PCR results are overall, congruent (Table 2). *Rickettsiella*, an intracellular endosymbiont of arthropods (*Leclerque, 2008*), was found in a tick from Ohio (1/64; tick ID 119572D) with a relative abundance of 11%.

## Statistical analysis: integrating individual data sets
### Clustering approach

The dendrogram resulting from hierarchical clustering showed that the two most distinct samples (clustering together) were from the same location and collection event: Fernald

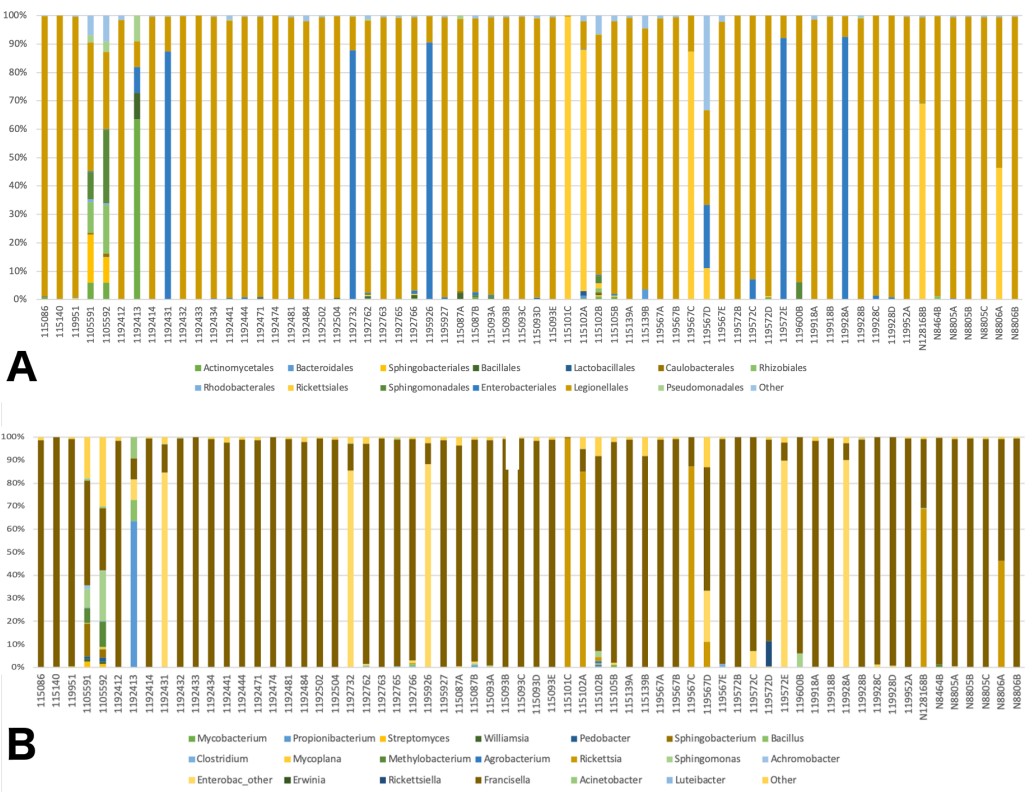

**Figure 2  Barplots of the relative abundance of bacteria.** (A) Barplot of the relative abundance of bacteria at the Order level. The *x* axis corresponds to individual tick specimens, and the *y* axis shows the relative abundance of the different bacteria Orders. Bacteria orders are color-coded as shown below the plot; (B) Similar to (A), barplot of the relative abundance of the bacteria at the genus level.

Preserve, Hamilton Co., Ohio (110559-1 and 110559-2) (Fig. 3). The second clade grouped two samples from CA, the two samples collected in Lake Co (115102A,B). The next pair of samples corresponded to the two remaining samples from the west coast, a sample from Napa Co, CA (115101C); and one from Whitman Co, WA (115105B). These two samples were closely associated with a clade that included all Northern samples. Lastly, all Eastern samples clustered together, without any internal grouping according to geography (Fig. 3).

No differences were evident upon visualization of community membership or structure. PCA did not show any specific clustering pattern between samples (File S2). In fact, most of the ticks were clustered in a single aggregate. The ticks that clustered somewhat separated were firstly the two that also appeared separated in the clustering analysis (110559-1 and 110559-2); and a tick from Ohio, 119241-3. It is important to note that the first two principal components explained only approximately 40% of the variance present in our sample.

As for $k$, the optimal number of clusters, silhouette method and elbow method suggested $k = 2$ and $k = 4$ respectively (File S3). Both values were close to the number of genetic clusters, which was also the number of clusters we used in the analysis. The C-index

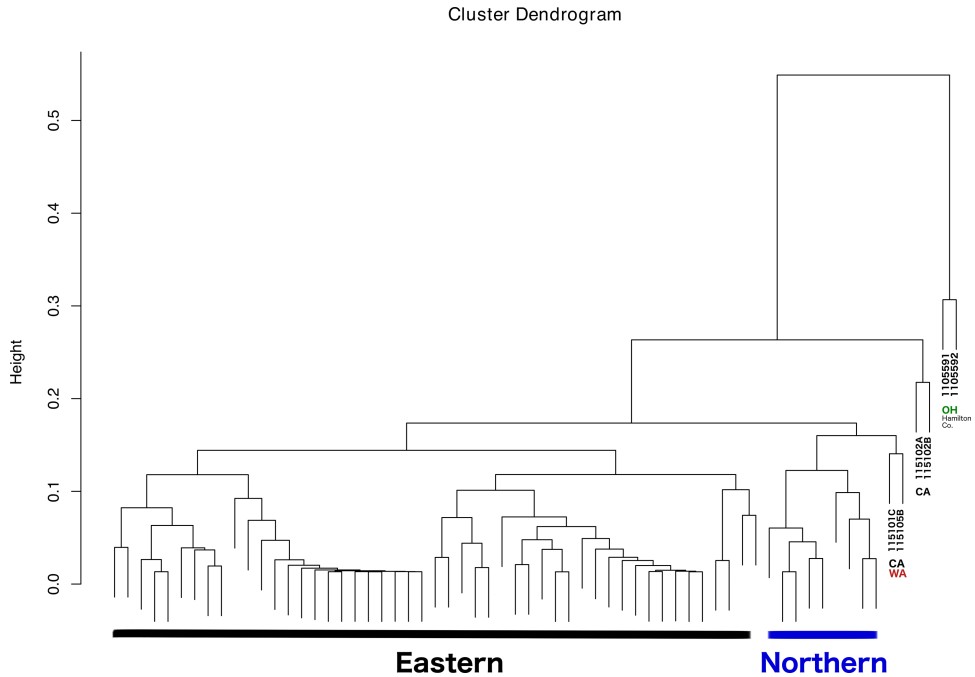

**Figure 3 Dendrogram showing the relationship between all samples, and the different groupings.** Groupings names: Eastern, Western, CA (samples from California), WA (samples from Washington), and OH Hamilton Co (samples collected in Hamilton County, Ohio).

between the predicted clusters and genetic clusters was 0.61, indicating some weak but nonnegligible consistency between these two pairs.

### Ordination methods

When visualizing the samples in the space through NMDS (stress value 0.064), the samples were spread out without any specific pattern (Fig. 4A). In PCoA with Gower's distance as metric, most samples appeared in a single tight aggregate, while a few others dispersed randomly along the axes (Fig. 4B).

### Test for significance for different variables.

The number of bacterial genera found in ticks belonging to the Eastern genetic cluster was higher than that of the Western, and Northern clusters. The genus *Francisella* was dominant in almost all the ticks, regardless of geography. Its relative abundance did not differ significantly between males and females (p-val 0.8); nor between ticks belonging to different genetic clusters (Eastern vs Western p-val 0.57, Eastern vs Northern p-val 0.96, Northern vs Western p-val 0.60) (Fig. 5). The relative abundance of this genus was also not significantly different between infected and uninfected ticks (p-val 0.14), although it was generally lower in infected ticks. Infected ticks were defined as those that were PCR positive for either *Anaplasma* or *Rickettsia* species.

*Rickettsia* relative abundance did not significantly differ between males and females (p-val 0.44), nor between ticks belonging to different genetic clusters (Northern vs. Eastern

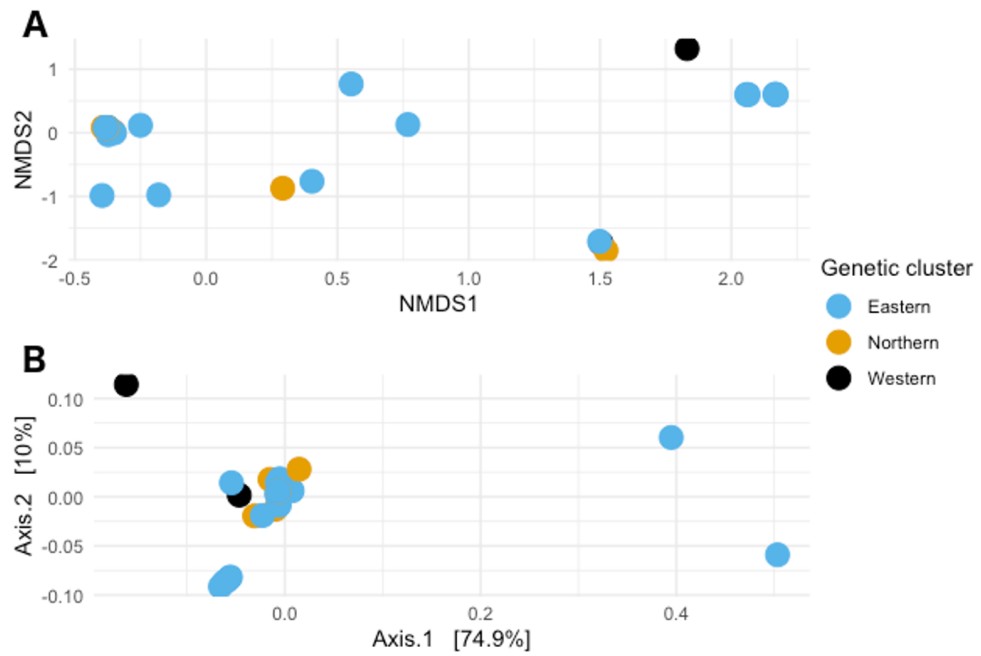

**Figure 4** **Non-metric multidimensional scaling and Principal coordinates analyses.** (A) Non metric multidimensional scaling analysis of the samples. Different colors represent the different genetic clusters: Eastern (red), Northern (green), and Western (blue). (B) Principal coordinates analysis of the samples. Different colors represent the different genetic clusters: Eastern (blue), Northern (orange), and Western (black). Axis 1 explains 74.9% of the variance, and Axis 2 explains 10% of it.

$p$-val $= 0.26$; Northern vs. Western $p$-val $= 0.72$; and Eastern vs. Western $p$-val $= 0.45$) (Fig. 5). *Rickettsia rhipicephali* and *Anaplasma* spp. were present only in ticks belonging to the Eastern genetic cluster, and *R. montanensis* was present in ticks belonging to the Northern and Eastern clusters. *Rickettsia bellii* was found infecting ticks from the Western and Northern genetic clusters, although of these ticks were collected in the Western region (Fig. 6). One of the ticks positive for this species was 115101C, a tick collected in California but with the genetic make-up corresponding to Northern ticks.

Overall, microbial beta diversity differed between regions ($p$-val $< 0.05$), but not between sexes ($p = 0.63$) when using Bray–Curtis distance. Post hoc tests indicated that ticks collected in the Western region harbored a different microbiome from those collected in the Eastern or Northern regions ($p < 0.05$). In the case of presence/absence of Bacteria (Jaccard distance), the results were the same. A significant difference between ticks collected from different regions was detected ($p < 0.05$), and the post hoc tests showed that the differences were significant between the Western and both the Eastern and Northern regions.

The Adonis test for genetic clusters showed that there were differences ($p < 0.05$) only when considering relative abundance of bacterial taxa (Bray–Curtis distance). When the pairwise Adonis test was performed to determine what genetic clusters differed in microbiome structure ($p < 0.05$) between Eastern and Western genetic clusters, but its

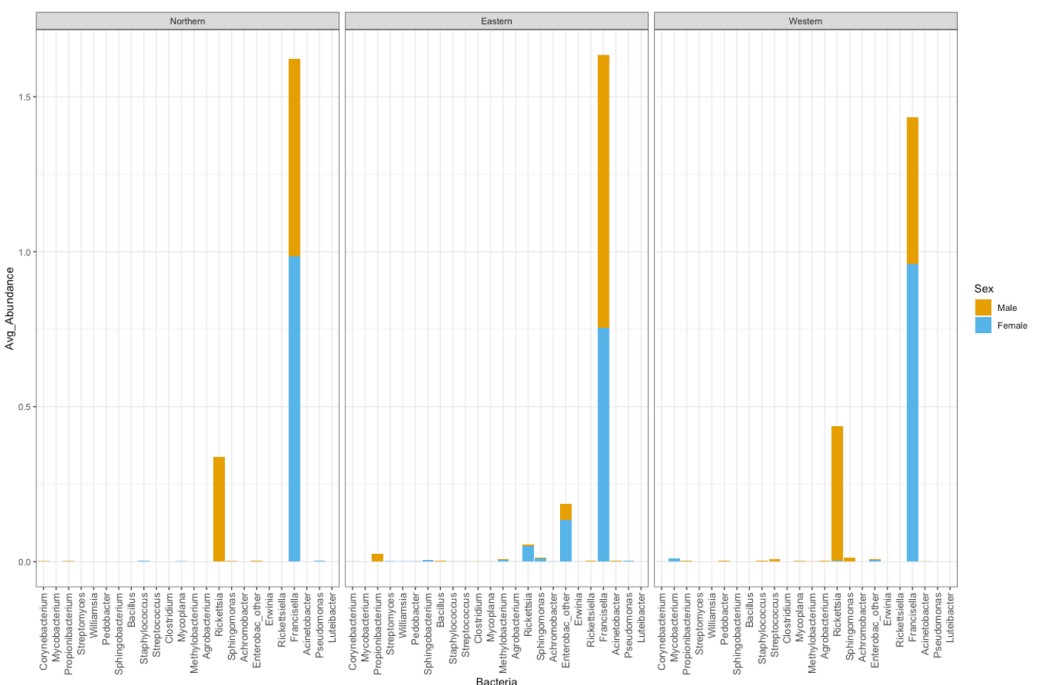

**Figure 5    Average abundance of bacteria genera by genetic cluster.** Average abundance ($y$ axis) of bacteria genera ($x$ axis) by genetic clusters (Northern, Eastern, Western), and sex (male in orange, female in blue).

correction was not (plFDR 0.096). Eastern and Western clades are genetically, the two more distant (*Lado et al., 2019*).

## DISCUSSION

This work corresponds, to the best of our knowledge, to the first attempt in tick research to integrate population genetic, microbiome, and pathogens presence data to better understand the ecology of TBDs. To optimize comparisons, it was our goal to generate those three pieces of data for each of the individual ticks included in the data set. We show that this is possible, even when techniques based on next generation sequencing require high quantities of high quality DNA.

We developed an effective pre-processing and processing procedure for researchers interested in microbiomes of small organisms, or parts of organisms with limited amounts of genetic material. This was motivated by a need for consensus regarding the pre-processing of ticks for generating microbiome data. For example, it is likely that many of the incongruencies between microbiome studies arise from differences in approaches to decontamination of the ticks' surface. Several studies in the literature performed surface sterilization of the ticks before DNA extraction (e.g., *Lado et al., 2018*; *Trout Fryxell & DeBruyn, 2016*), whereas others did not (e.g., *Clow et al., 2018*); and the differences in bacterial communities may be a reflection of extensive environmental "contamination" in the latter. Even between studies that "washed" the ticks, comparisons need to be done

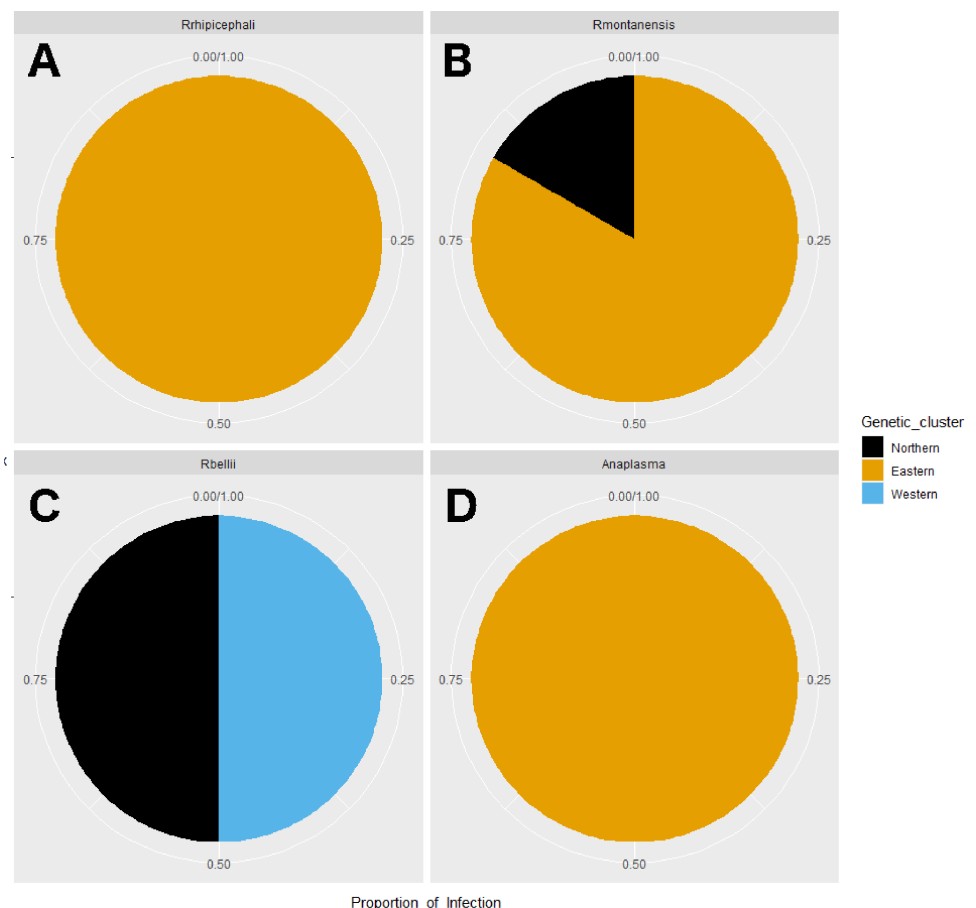

**Figure 6** **Pie chart plots of proportions of infection by genetic clusters.** Each plot corresponds to an agent; (A) *R. rhipicephali*, (B) *R. montanensis*, (C) *R. bellii*, (D) *Anaplasma* spp. The colors represent the genetic clusters (Northern in black; Eastern in orange; and Western in blue).

cautiously, since not all sterilization methods are equally effective. The incorporation of bleach appears necessary for effective decontamination (*Binetruy et al., 2019*). Increased consistency in results of microbiome studies of ticks requires some community consensus related to the generation and analysis of microbiome data. This is especially important for non-model organisms such as ticks, given the "noise" (Alpha and Beta diversity inflation) generated by the presence of environmental microorganisms.

Consistent with most other studies of tick microbiomes (*Van Treuren et al., 2015*; *Gall et al., 2016*; *Chicana et al., 2019*), our analysis showed that the microbiome of many species is heavily dominated by a few genera, while the microbiomes of others, such as *Ixodes angustus* (*Chicana et al., 2019*) and *Haemaphysalis lemuris* (*Lado et al., 2018*), are more diverse. The microbiome of *D. variabilis* is dominated by *Francisella* (*Chicana et al., 2019*; *Travanty et al., 2019*; current study). *Clow et al. (2018)* presented, to some extent, opposing results (far higher microbial diversity), but they analyzed only 9 specimens, and the ticks were not surface sterilized. These results casts some doubt on the concept of a "core

microbiome". For example, *Chicana et al. (2019)* report a *D. variabilis* core microbiome as composed by *Francisella*, *Sphingomonas*, and *Methylobacterium*. On the other hand, *Travanty et al. (2019)* reported a *D. variabilis* core microbiome that includes *Francisella* spp., *Sphingomonas* spp., *Delftia* spp., and *Hymenobacter* spp. Our study recovered all of these taxa but only *Fransicella* spp. is nearly universally present. *Francisella* has been well established as endosymbiont and dominant in *D. variabilis*, as well as other *Dermacentor* species. All other taxa reported as "core microbiome" are not consistent across studies (e.g., *Chicana et al., 2019*; *Clow et al., 2018*; *Rynkiewicz et al., 2015*; *Travanty et al., 2019*). Thus, the true "core" of taxa overlapping across studies, can be reduced to *Francisella* only. The concept of "core microbiome" is generally used in microbiome research to refer to a suite of microbes, and not to refer to only one taxon. The utility of the concept of "core microbiome" for *D. variabilis* is therefore unclear.

Recent studies focusing on the pathogen transmission by *D. variabis* and on its role as vector of human diseases have consistently reported low prevalence of infections with known pathogenic microorganisms. For example, *R. rickettsii*, is usually present in ~1% of the ticks analyzed e.g., (*Hecht et al., 2019*). The results obtained in the present study are consistent with these literature results, and show an overall *Rickettsia* spp. prevalence of 17.2%, including *R. montanensis, R. bellii,* and *R. amblyommatis* infections. *Rickettsia rickettsii* DNA was not found in any of the ticks analyzed. We detected DNA of *Anaplasma* spp. in one tick sample collected in Oklahoma. Its DNA sequence matched 100% with GenBank sequences submitted by researchers at the CDC Fort Collins, who isolated the agent from a human blood sample. No additional information is available in the literature about this case. This study comprises the first report of that specific bacterial agent in *D. variabilis*. However, an *Anaplasma* spp. *bovis*-like agent has been previously reported in *D. andersoni* ticks from Canada (*Dergousoff & Chilton, 2011*; *Chilton, Dergousoff & Lysyk, 2018*). Unfortunately, the region of 16S amplified differs between our study and that of *D. andersoni* (*Dergousoff & Chilton, 2011*), making it difficult to determine if the sequences correspond to the same specific agent. Regardless, it appears that uncharacterized *Anaplasma* agents are circulating in nature (ticks and humans), and their characterization, together with the determination of whether or not they are capable of causing disease in humans should be further explored. Furthermore, due to the finding of this agent's DNA in a *D. variabilis* sample in this study, the role of *D. variabilis* as potential vector should be considered and further explored.

With respect to the primary goal of this preliminary study, the integration of the different types of data was done successfully, although the interpretation of the results is somewhat challenging. The latter problem is most likely caused by to the low levels of genetic diversity and moderate levels of population structure (see details in *Lado et al., 2019*), and a microbiome highly dominated by a few taxa. As a result, ordination methods failed to show patterns of variation across different groups. Nonetheless, the dendrogram resulting from the clustering analysis, in which all lines of evidence and geographic location were considered, was largely congruent with the observations at the population genetic level. Thus, the addition of microbiome, pathogen presence, ticks sex, and geographic location led to conclusions that are, in general, consistent with ticks genetics. All Eastern

samples clustered together (with the exception of two samples from Hamilton Co, OH), and separated from both Western and Northern samples. The distinctiveness and separation of those two samples from Hamilton Co, OH reflects their distinctiveness at the microbiome level (Fig. 2). The fact that both samples appeared very similar to all other Eastern samples at the population genetic level (*Lado et al., 2019*), supports that hypothesis. It is possible that these two specimens were not well surface sterilized, or that the environment at that collection location is different.

The higher number of bacterial genera found in ticks belonging to the Eastern genetic cluster when compared to either Northern or Western ticks, is likely a product of the higher number of ticks analyzed, and the greater geographic area covered. Even though in the broader sense the microbiome is not diverse (dominated by three genera), it differed between the three main geographic regions: Eastern, Western, and Northern. Statistical results looking at the beta diversity between genetic clusters are less clear: initial test demonstrated a difference in the microbial communities between genetic clusters, but pairwise tests with their corresponding corrections failed to reach the same result. This may be a result of one of the study's limitations, the number of samples. Thus, these findings should be further explored using more samples, and more samples per genetic cluster. From our preliminary analyses, geography explains the differences in the microbial communities better than host genetics. This could be the result of certain microorganisms occurring only in certain geographic areas.

Despite challenges arising during the interpretation of the results, the integration of lines of evidence and metadata, revealed that Eastern ticks can be separated from Northern and Western ticks. It can also be noted that some *Rickettsia* species were associated with a certain geographic area. For example, *R. bellii* was found only in samples collected in CA, a finding consistent with that reported in *Hecht et al. (2019)*. It seems possible that *R. bellii* is more common along the west coast, although more samples should be analyzed to confirm this. And while the two *R. bellii* positive tick samples were collected in CA, one of them (ID 115101C) did not belong to the Western genetic cluster; it belonged to the Northern cluster. From the host genetics perspective, this tick is more similar to ticks from Northern locations; whereas from the rickettsial agents perspective, it is more similar to Western samples. Looking at the microbiome composition, this tick could belong to any geographic region. This underscores the value of integrating different types of data when thinking about disease ecology.

The approaches taken during this study, both to generate and analyze data, can be applied to a wide variety of taxa, and groups of organisms. It is likely that in organisms with a higher level of variation, the clustering methods will be more informative, and their interpretation more straightforward. Therefore, the approaches used herein have potential, and could greatly improve future studies looking at different aspects of diseases ecology. In the particular case of *D. variabilis,* the approaches used herein failed to detect clear tendencies or patterns. This was likely due to the low levels of variation (both in the microbiome and population genetics), and the very low prevalence of pathogenic microorganisms.

## CONCLUSIONS

In conclusion, this is the first study that successfully generated microbiome, population genetics, and pathogens presence data for the same individual ticks. General methodologies and pre-processing steps are replicable, and applicable to different groups of organisms across the tree of life. This work also comprises, at present, one of the few studies aiming at integrating population genetics and microbiome data to better understand ecological processes and disease, and it is the first one to do so for ticks. The integration of different lines of evidence allows a more holistic approach; and clustering and ordination methods are very helpful to summarize and visualize the results. Finally, this study comprises the first report of "Uncultured *Anaplasma* sp. clone 15-3642" in *D. variabilis*. This agent was previously isolated from human blood, and it is important to determine if it is capable of causing disease in humans, and if so, the role of *Dermacentor* ticks as potential vectors.

## ACKNOWLEDGEMENTS

We are grateful to M. Yoshimizu, A. Donohue, B. Ryan, G. Keeney, C. Nelson, L. Beati, L. Durden, C. Lubekzyk, D. Neitzel, G. Hickling, and B. Pagac who provided some of the tick specimens used in this study. We also thank The National Ecological Observatory Network, a program sponsored by the National Science Foundation and operated under cooperative agreement by the Battelle Memorial Institute, for providing some specimens used in this work. We thank the Ohio Supercomputer Center for computing resources, and two anonymous reviewers for their constructive comments, which helped improving earlier versions of this manuscript. The findings described herein are those of the authors and do not necessarily represent the official position of the United States Department of Health and Human Services.

### Funding

This work was supported by the Ohio State University: the Graduate School's Alumni Grants for Graduate Research and Scholarship to Paula Lado. There was no additional external funding received for this study. The funders had no role in study design, data collection and analysis, decision to publish, or preparation of the manuscript.

### Grant Disclosures

The following grant information was disclosed by the authors:
Ohio State University: the Graduate School's Alumni Grants for Graduate Research and Scholarship.

### Competing Interests

The authors declare there are no competing interests.

## Author Contributions

- Paula Lado, Bo Luan conceived and designed the experiments, performed the experiments, analyzed the data, prepared figures and/or tables, authored or reviewed drafts of the paper, and approved the final draft.
- Michelle E.J. Allerdice conceived and designed the experiments, performed the experiments, authored or reviewed drafts of the paper, and approved the final draft.
- Christopher D. Paddock, Sandor E. Karpathy and Hans Klompen conceived and designed the experiments, authored or reviewed drafts of the paper, and approved the final draft.

## Field Study Permissions

The following information was supplied relating to field study approvals (i.e., approving body and any reference numbers):

Metro Parks permitted collection of tick species throughout the Metro Parks. The Division of State Parks (DNR), Indiana Department of Natural Resources authorized tick collection at Brown County State Park, Clifty Falls State Park, O'Bannon Woods State Park, Monroe Lake, Splinter Ridge Fish and Wildlife Area, Clark State Forest, and Ferdinand State Forest.

## Data Availability

The data are available under NCBI SRA: PRJNA606891.

## Supplemental Information

Supplemental information for this article can be found online at http://dx.doi.org/10.7717/peerj.9367#supplemental-information.

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
