# Peer review of "Integrating population genetic structure, microbiome, and pathogens presence data in Dermacentor variabilis"

_PeerJ, doi:10.7717/peerj.9367_

## Round 0.1 · original submission · Major Revisions

Dear Dr. Lado and colleagues:

Thanks for submitting your manuscript to PeerJ. I have now received two independent reviews of your work (I was waiting on a third that never came, my apologies), and as you will see, the reviewers raised some concerns about the research. Despite this, this the reviewers are optimistic about your work and the potential impact it will have on research studying Dermacentor variabilis population genetic structure, microbiome and pathogen vector capacity. Thus, I encourage you to revise your manuscript, accordingly, taking into account all of the concerns raised by both reviewers.

I agree with many of the concerns of the reviewers, and thus feel that their suggestions should be adequately addressed before moving forward. Please understand the limitations of sample size in microbiome studies, specifically what can and cannot be inferred based on sample variance and geographic coverage.

Your sample size is laudable, and I think the study has generated meaningful data from the same ticks, something not done all that much in the vector biology literature. I would stress spending some time summarizing the overall meaning of this combination of data and specifically conclude whether it is worth doing in future survey studies of this nature.

Please note that Reviewer 2 kindly provided a marked-up version of your manuscript.

I look forward to seeing your revision, and thanks again for submitting your work to PeerJ.

Good luck with your revision,

-joe

Reviewer 1 ·

Basic reporting

Some word spelling, grammar and writing format should be checked throughout the manuscript. Specific points:
Line 387, “(Dergousoff and Chilton, 2011)” should be changed to “Dergousoff & Chilton, 2011” according to the requirement of the Journal.
Line 421, “Norther” should be changed to “Northern”
Line 265 and 285, please confirm the writing form.
Please unify the format of the titles at all levels, and carefully double checked the references of the manuscript.

Experimental design

no comment

Validity of the findings

no comment

Additional comments

Lado et al. studied the intergration of population genetic structure, microbiome, and pathogens presence data for Dermacentor variabilis. This research is of great significance for tick-borne diseases. And the following is my comments.

1. In this study, the reason for choosing Dermacentor variabilis and a brief introduction of this species should be given in the part of “Introduction”.

2. From the figure 1, the number and distribution of samples are relatively fewer. Are the selected sampling localities representative?

3. Whether need to consider the influence of sex and developmental stages in the present study. And why the authors choose to detect the Rickettsiales, not other pathogens?

4. Line 91-95, “In amphibians, Griffiths et al. (2018) showed that at the landscape level, the microbiome was not structured according to host population structure even though the genetic distance among hosts was still significantly correlated to microbial community dissimilarity when controlling for geographic distance.” This sentence is not very clear, please rewrite it.

5. Line 400-402, “Thus, the addition of microbiome, pathogen presence, host sex, and geographic location led to conclusions that are, in general, consistent with host genetics”. Host information or references were not mentioned in this part, and how is this conclusion drawn?

Reviewer 2 ·

Basic reporting

There is no scientific question in this article. The authors should clearly define their objectives and their scientific question.
I strongly advice the authors to increase the number ot tick samples or focus their study on the tick microbiome analysis.

Experimental design

There is no scientific question in this article.
The number of samples is too low. Authors cannot reasonably perform population genetic and pathogen analysis on only 64 ticks collected in 12 different sates (tick pathogen analysis regularly performed on only one tick).

Validity of the findings

The number of samples is too low to perform this type of studies.
No negative controls for the tick microbiome analysis.

Additional comments

In this study, authors generated microbiome and pathogen presence data for Dermacentor variabilis, and integrated those data sets with population genetic data, and metadata for the same individual tick specimens. The total number of ticks collected in 12 states was 64 ticks. Clustering and multivariate statistical methods were used to combine, analyze, and visualize data sets. As mentioned by the authors, interpretation of the results is challenging, likely due to the low levels of genetic diversity and the high abundance of a few taxa in the microbiome, and to my opinion the too low number of samples. From the tick microbiome analysis, they found that Francisella was dominant in almost all ticks, regardless of geography or sex. Nevertheless, results showed that, overall, ticks from different geographic regions differ in their microbiome composition. Additionally, DNA of Rickettsia rhipicephali, R. montanensis, R. bellii, and Anaplasma spp., was detected in D. variabilis specimens. Finally, authors summarize their study only saying that « this is the first study that successfully generated microbiome, population genetics, and pathogen presence data from the same individual ticks, and that attempted to combine the different lines of evidence. »

Authors are right saying that interdisciplinary approaches are essential and required to improve our understanding of TBP ecology. The integration of these different approaches is the key to improve our understanding. However, I did not find this data integration in the paper and the authors do absoultely not answer and hypothesize to this potential link between tick population genetic, tick-borne pathogens and microbiomes. The objectives are unclear. No scientific questions emerge from this study. The number of samples is too low to perform this type of integrated studies and obtain accurate data for the analysis of tick genetic population and pathogen prevalences. At no time authors discuss about these limitations. In my opinion, this study cannot be considered for publication and I strongly advice the authors to increase the number ot tick samples or focus their study on the tick microbiome analysis.

Major :
- There is no scientific question in this article.
What is the scientific question in this study ? The authors mentioned L.36-38. « This is the first study that successfully generated microbiome, population genetics, and pathogen presence data from the same individual ticks, and that attempted to combine the different lines of evidence ». Combining different approaches in a study is not a scientific question. What is the goal of this study ? For each approach, authors have descriptive results but no links between these different results appear in the study. Authors are completly right when they say that « we still do not fully understand how host genetics and environmental factors interact to shape the microbiome of organisms, or how pathogenic microorganisms affect the microbiome and vice versa. They are still right when they say that « the integration of different lines of evidence may be the key to improve our understanding of TBDs ecology. ». These data and their combined analysis are crucial to keep improving our knowledge on TBD ecology. With such a context in the introduction, I expected that this study would provide new results and hypothesis. After reading the entire article, I did not find any accurate and relevant information about that.
The authors should clearly define their objectives and their scientific question.

- The number of samples is too low to perform this type of studies.
Authors cannot reasonably perform population genetic and pathogen analysis on only 64 ticks collected in 12 different sates (in three states, only one tick was collected). 8 states have a number of collected specimen < 5. This is unbelivably low to (i) have an accurate estimation of pathogen prevalences, (ii) identify and compare tick microbiome composition and diversity and (iii) perform robust and significant statistical analyses. For three of these 8 states, authors collected only one tick ; One (in North Dakota) was positive for Rickettsia montanensis… so authors concluded that the prevalence was 100%. This conclusion does not make sens. Authors need to have more ticks to estimate the pathogen prevalences (with standard deviations…).
Performing this type of studies and integrating the different approaches needs to collect and analyze much more ticks to have a more accurate estimation of both pathogen prevalences and population genetic. 64 ticks in 12 states (almost 50% only in Ohio) is too low.
The authors mentioned in the abstract : « Interpretation of the results is challenging, likely due to the low levels of genetic diversity and the high abundance of a few taxa in the microbiome ». I guess the low number of samples is probably another reason.


- Negative controls
L177-194 : Concerning the tick microbiome analysis, authors did not mention if they performed negative controls ? Several negative controls have to be done at each step of the process (crushing, extraction, amplification…). These controls have then to be sequenced to exclude contaminant sequences from the final dataset. In low biomass samples, (i.e. individual tick samples), these contaminants may represent a large part of the obtained sequences, and thus generate considerable errors in downstream analyses and in the result interpretation. Please provide detailed information about the controls you performed in this study.

Minor :
L.42-111 : The introduction part is too long. The part from L.69-79 is, in my opinion, not essential. Please shorten the introduction and get straight to the point stating clearly the objectives of this study.

L. 117 : 64 adults : Please provide information about the number of males and females, their localization. Authors cana dd these information in the Table 1.

L.198-199 : What do you mean by « the dimensions of the data were therefore substantially reduced. »

L.260-264 : What about R2 (R) ? Did you merge your sequences ? With your primers, you should obtain amplicons around 400-420bp. You mentioned 251bp. I do not understand these results.

L.263-264 : « the average number of sequences per tick specimen included in downstream analyses was fairly low, ranging from a few hundred to ~15,500 ». The number of sequences was strongly different between samples. Did the authors homogeneize this number of sequences to obtein an equal number in each sample ? This step is essential to compare samples. Please add this information in the M&M part.

L.272 : Rickettsiella is also a tick endosymbiont. Please improve this sentence with some other newer references.

L.297-299 : Please provide the stress value R of the NMDS. An NMDS ordination with a stress value around or above 0.2 is deemed suspect and a stress value approaching 0.3 indicates that the ordination is arbitrary. Stress values equal to or below 0.1 are considered fair, while values equal to or below 0.05 indicate good fit.

Annotated reviews are not available for download in order to protect the identity of reviewers who chose to remain anonymous.

---

## Round 0.2 · Minor Revisions

Dear Dr. Lado and colleagues:

Thanks for resubmitting your manuscript to PeerJ. I have now received two reviews of your work, and as you will see, these two reviewers still raised some concerns about the research. Reviewer has identified a few minor errors in the writing. Reviewer 2 still has several issues with the analyses and data availability. Please also address all of these issues raised by the two reviewers.

Therefore, I am recommending that you revise your manuscript, accordingly, taking into account all of the issues raised by the reviewers. I do believe your manuscript will be closer to publication once these issues are addressed.

I look forward to seeing your revision, and thanks again for submitting your work to PeerJ.

Good luck with your revision,

-joe

Reviewer 1 ·

Basic reporting

Some writing format should be checked throughout the manuscript. Take some lines for example: Line 117, 286, 306, 311.
Besides, there are so many mistakes in the writing of the references in the MS. For example: Line 482, “12(268)” should be changed to “12: 268”
Line 582, “PLoS ONE 8” should be changed to “PLoS ONE 8:e61217”.
Line 617, “Genome Biology and Evolution 7” should be changed to “Genome Biology and Evolution 7(3):831–838”.
Line 627, “1-19” should be changed to “e0155559”.
More, I won’t list the other mistakes in the part of “References”, please check carefully according to the Journal Guideline.

Experimental design

no comment

Validity of the findings

no comment

Reviewer 2 ·

Basic reporting

the authors’ have not given raw data, i.e. how many total sequences were there, how many sequences per sample before filtering, how many sequences before and after quality control (QC), mean +/- and OTU +.- per sample. This is very necessary information. Information about sequences detected in negative controls are also crucial. Please provide in the supplemental material an Excel file with the OTU tables (before releasing the OTUs detected in controls and after releasing these OTUs).
Do the authors submitted their sequences to GenBank ? please provide the Accession Numbers !

Experimental design

I understood the authors considered ticks per « regions » (East, West and North). If I am right (based on the size of circles on the figure 1), the authors have considered the « western cluster » with only four specimens (this is an hypothesis because all these information are missing in the ms, please provide the number of specimens per region). I reiterate my comment : for me, the number of samples is unbelivably low to (i) have an accurate estimation of pathogen prevalences, (ii) identify and compare tick microbiome composition and diversity and (iii) perform robust and significant statistical analyses. Estimation of pathogen prevalences need standard deviations, please provide these information in the table 2.
A part of the author answer is : « As for the low prevalence, unfortunately it is very low. And this would not be fixed by increasing the number of ticks by a hundred… this is the reality ». Your answer means that you can identify and generalize the tick microbiome and the prevalence of tick-borne pathogens in the West coast analyzing only four ticks. I am sorry, I am not convinced by this answer at all. All the epidemiologists around the world try to get the higher number of specimens to better estimate the prevalence of pathogens and the authors assure that increasing the number of ticks is not necessary and only few specimens is needed. If you really think that this answer is correct and relevant, please write very rapidly an opinion paper on this : this future paper should interest all the epidemiologists.

Validity of the findings

The authors mention in their answer that the reads did not merge well. Could they include this information in the MS and try to explain this result? In my opinion, this could suggest that the quality of the sequencing and/or the preparation of the amplicon library were not optimal. Could the authors provide the sequencing quality controls? What was the range (number of bp) the authors used for merging reads ?
Concerning my remark on the lack of negative controls, the authors only mentioned they used two negative controls. Because these negative controls are crucial for the next steps of the sequence analyses, we need much more information about these controls… What kind of controls ? At which steps these controls have been performed ? please provide details on these controls.

Additional comments

I thank the authors for their answers. I am sorry but I have still some remarks/questions :
I understood the authors considered ticks per « regions » (East, West and North). If I am right (based on the size of circles on the figure 1), the authors have considered the « western cluster » with only four specimens (this is an hypothesis because all these information are missing in the ms, please provide the number of specimens per region). I reiterate my comment : for me, the number of samples is unbelivably low to (i) have an accurate estimation of pathogen prevalences, (ii) identify and compare tick microbiome composition and diversity and (iii) perform robust and significant statistical analyses. Estimation of pathogen prevalences need standard deviations, please provide these information in the table 2. A part of the author answer is : « As for the low prevalence, unfortunately it is very low. And this would not be fixed by increasing the number of ticks by a hundred… this is the reality ». Your answer means that you can identify and generalize the tick microbiome and the prevalence of tick-borne pathogens in the West coast analyzing only four ticks. I am sorry, I am not convinced by this answer at all. All the epidemiologists around the world try to get the higher number of specimens to better estimate the prevalence of pathogens and the authors assure that increasing the number of ticks is not necessary and only few specimens is needed. If you really think that this answer is correct and relevant, please write very rapidly an opinion paper on this : this future paper should interest all the epidemiologists.
The authors mention in their answer that the reads did not merge well. Could they include this information in the MS and try to explain this result? In my opinion, this could suggest that the quality of the sequencing and/or the preparation of the amplicon library were not optimal. Could the authors provide the sequencing quality controls? What was the range (number of bp) the authors used for merging reads ?
Concerning my remark on the lack of negative controls, the authors only mentioned they used two negative controls. Because these negative controls are crucial for the next steps of the sequence analyses, we need much more information about these controls… What kind of controls ? At which steps these controls have been performed ? please provide details on these controls.
Finally, the authors’ have not given raw data, i.e. how many total sequences were there, how many sequences per sample before filtering, how many sequences before and after quality control (QC), mean +/- and OTU +.- per sample. This is very necessary information. Information about sequences detected in negative controls are also crucial. Please provide in the supplemental material an Excel file with the OTU tables (before releasing the OTUs detected in controls and after releasing these OTUs).
Do the authors submitted their sequences to GenBank ? please provide the Accession Numbers !

---

## Round 0.3 · accepted · Accept

Dear Dr. Lado and colleagues:

Thanks for revising your manuscript based on the concerns raised by the reviewers. I now believe that your manuscript is suitable for publication. Congratulations! I look forward to seeing this work in print, and I anticipate it being an important resource for groups studying Dermacentor variabilis population genetic structure, microbiome and pathogen vector capacity. Thanks again for choosing PeerJ to publish such important work.

Best,

-joe

Reviewer 1 ·

Basic reporting

no comment

Experimental design

no comment

Validity of the findings

no comment